# Neuroimaging of Cryptococcal Meningitis in Patients without Human Immunodeficiency Virus: Data from a Multi-Center Cohort Study

**DOI:** 10.3390/jof9050594

**Published:** 2023-05-19

**Authors:** Seher H. Anjum, John E. Bennett, Owen Dean, Kieren A. Marr, Dima A. Hammoud, Peter R. Williamson

**Affiliations:** 1Laboratory of Clinical Immunology and Microbiology (LCIM), National Institute of Allergy and Infectious Diseases (NIAID), National Institutes of Health (NIH), Bethesda, MD 20892, USA; 2Department of Dermatology, School of Medicine and Dentistry, University of Minnesota, Minneapolis, MN 55455, USA; 3Department of Medicine, Johns Hopkins University, Baltimore, MD 21205, USA; 4Center for Infectious Disease Imaging (CIDI), Radiology and Imaging Sciences, Clinical Center, National Institutes of Health, Bethesda, MD 20892, USA

**Keywords:** neuroimaging, MRI, cryptococcus, meningitis, HIV-negative

## Abstract

Background: A clearer understanding is needed about the use of brain MRI in non-HIV patients with cryptococcal meningitis. Methods: Cerebral CT and MRI were studied in 62 patients in a multicenter study of cryptococcal meningitis in non-HIV patients. CT was performed in 51 and MRI in 44. MRI results are reported for the images read at NIH for 29 of the 44 patients. CT reports obtained from the original REDCap database were added to calculate the incidence of normal findings. Results: CTs were read as normal in 24 of 51 (47%), MRIs were normal in 10% (three of 29). The most characteristic lesions of cryptococcal meningitis on MRI were small basal ganglia lesions representing dilated perivascular spaces in 24% and basal ganglia lesions with restricted diffusion (infarcts) in 38%. In the 18 patients who received contrast, contrast-enhancing lesions, likely representing masses of cryptococci and inflammatory cells, were found in the basal ganglia in 22% and elsewhere in the brain in 22%. Meningeal enhancement was seen in 56%, ependymal enhancement in 24%, and choroid plexus enhancement in 11%. Hydrocephalus was found in five (18%), though increased intacranial pressure was not detected. Suboptimal imaging (n = 6), lack of contrast administration (*n* = 11) and lack of follow-up, however, markedly limited the accurate assessment of abnormalities in multiple cases. Conclusion: MRI characteristics of non-HIV cryptococcal meningitis include hydrocephalus, meningeal and ependymal enhancement and basal ganglia lesions. Optimal imaging is, however, necessary to maximize the diagnostic and prognostic usefulness of MRI.

## 1. Introduction

Central nervous system cryptococcal meningitis (CM) is a major cause of morbidity and mortality in HIV co-infected individuals but is also an important cause of disease in patients on a range of immunosuppressive regimens as well as previously healthy individuals without obvious immunosuppression [1]. Mortality remains high in all populations, suggesting a need to better understand the pathophysiology of the disease in ways that may improve outcome in individual patients. Besides basic structural information, neuroimaging of the brain has improved our understanding of the neuropathology of cryptococcal meningitis. For example, diffusion weighted imaging (DWI) with apparent diffusion coefficient maps (ADC) can identify foci of acute and subacute infarcts, while contrast-enhanced fluid attenuated inversion recovery (FLAIR) imaging can demonstrate subtle foci of meningeal enhancement. We here present a summary of published cerebral imaging results, with the caution that any summary is complicated by the variability of the immunocompromised populations in the “Non-HIV” category, differences in MRI instruments used at different centers and the frequent absence of definitions for the imaging terms used (Table 1). As shown in Table 1, results of MRI have differed somewhat between patients co-infected with HIV and those who are not, with infarcts, hydrocephalus and enhancement of the leptomeninges more commonly observed in the latter population. Most publications did not distinguish between HIV patients who were undergoing immune reconstitution but in those reports that did, meningeal enhancement was more common after antiretrovirals were begun [2]. Most infarcts were categorized as lacunar, meaning less than 15 mm diameter, and typically multiple. Cryptococcosis also causes parenchymal lesions in the brain, often called cryptococcomas, which are masses of encapsulated cryptococci with little or no surrounding inflammation. Also seen are enlarged Virchow–Robin (VR) spaces, often referred to as pseudocysts when seen as clusters of punctate lesions, reflecting perivascular spaces filled with masses of cryptococci. Cryptococcomas and enlarged VR spaces have been described in roughly equal proportions in HIV and non-HIV patients (Table 1). Dilated VR spaces are typically described in the basal ganglia. Atrophy is more commonly reported in HIV-infected patients but this can sometimes be confounded by patient age. Contrast enhancement of the choroid plexus and ependyma have been occasionally reported but not with sufficient frequency to compare non-HIV and HIV patients. However, we have reported that choroid plexitis and ependymitis were more prevalent in non-HIV-infected individuals than those co-infected with the virus [3]. In contrast to MRI, CT imaging of the brain has been reported to be abnormal in only about half the patients (Table 1) [4].

The present study reports our experience with MRI imaging of non-HIV patients with cryptococcal meningitis in 25 centers across the USA [5]. Images from 29 patients were examined by an experienced neuroradiologist (DAH) and described in defined terms.

**Table 1 jof-09-00594-t001:** Published imaging results on persons with cryptococcal meningitis.

HIV Status		Non-HIV			HIV			
Feature	Reference	#	Total	%	Reference	#	Total	%
Normal CT								
	Tjia [6]	13	25	52%	Charlier [4]	26	55	47%
	Tan [7]	10	20	50%	Tien [8]	9	29	31%
Normal MRI								
	Tan [9]	2	18	11%	Loyse [10]	2	87	2%
					Miszkiel [11]	4	25	16%
					Charlier [4]	2	24	8%
Meningeal enhancement								
	Lu [12]	8	15	53%	Miszkiel [11]	4	25	16%
	Tsai [13]	29	65	45%	Loyse [10]	24	87	28%
	Tan [9]	8	17	47%	Mathews [14]	1	5	20%
	Zhong [15]	71	114	62%	Xia [16]	21	55	38%
	Singh [17]	8	16	50%	Hammoud [3]	9	11	81%
	Hammoud [3]	32	45	71%				
Infarcts								
	Chen [18]	7	37	19%	Loyse [10]	12	87	14%
	Chang	7	12	58%	Charlier [4]	2	55	4%
	Tsai [13]	14	65	22%	Nguyen [19]	1	36	3%
	Nguyen [19]	3	24	13%				
	Zhong [15]	5	114	4%				
Hydrocephalus								
	Lu [12]	5	15	33%	Loyse [10]	2	87	2%
	Tsai [13]	21	65	32%	Charlier [4]	2	55	4%
	Tjia [6]	5	25	20%	Hammoud [3]	1	11	9%
	Lee [20]	10	76	13%				
	Zhong [15]	38	114	33%				
	Singh [17]	2	16	13%				
	Hammoud [3]	23	45	51%				
Cryptococcoma (other parenchymal lesions)								
	Tan [9]	11	18	61%	Tien [8]	3	10	30%
	Lee [20]	12	76	16%	Loyse [10]	23	87	26%
	Zhong [15]	35	114	31%	Miszkiel [11]	9	25	36%
	Singh [17]	6	16	38%	Charlier [4]	4	24	17%
	Hammoud [3]	22	45	49%	Hammoud [3]	3	11	27%
Dilated V-R spaces								
	Zhong [15]	45	114	39%	Mizskiel [11]	4	25	16%
	Tsai [13]	28	65	43%	Andreula [21]	5	9	56%
	Tan [9]	5	18	28%	Wehn [22]	2	2	100%
	Hammoud [3]	25	45	56%	Charlier [4]	11	24	46%
					Mathews [14]	5	15	33%
					Loyse [10]	31	87	36%
					Xia [16]	28	55	51%
					Hammoud [3]	5	11	45%
Atrophy								
	Zhong [15]	7	114	6%	Loyse [10]	30	87	34%
					Charlier [4]	4	24	17%
					Tien [8]	13	30	43%

## 2. Materials and Methods

Patients included in this report were part of a multi-institutional study (CINCH cohort) of 145 non-HIV patients with cryptococcosis (previously reported [5]). Neuroimaging results were not included in the prior report and are presented here. A detailed description of these cases was abstracted from details listed in a REDCap database. Under the CINCH protocol, people who developed cryptococcosis without HIV consented for longitudinal follow-up, under a protocol approved by the Johns Hopkins University Institutional Review Board (protocol NA_00051263). Of this cohort, we analyzed 65 patients who had proven cryptococcal meningitis, based on CSF culture positivity for *Cryptococcus neoformans* complex or a positive CSF antigen test. One culture was identified as *Cryptococcus gattii,* one of the species in the *C. neoformans* complex. Three patients were excluded from further analysis: 2 patients who never had imaging done and one who had a CT only on follow-up, leaving 62 for this report. 

### Available Imaging Studies

Of the 62 patients with imaging, 21 only had a CT scan of the brain on admission, 11 had an MRI and 30 had both. Adding initial studies conducted more than 15 days after admission, 18 had only a head CT, 11 had only an MRI and 33 had both. With the exception of a report of normal findings, imaging results of the CTs and MRIs in the REDCap database will not be analyzed in detail due to insufficient information. Of the 44 patients with MRI scans, 29 patients whose MRI images were sent to NIH were selected for the study. Only six of these 29 patients had follow-up MRIs, an insufficient sample to comment on resolution or persistence of abnormalities. 

In interpreting MRIs, foci of restricted diffusion in typical locations such as the basal ganglia were classified as definite or likely infarct when they showed expected signal (hyperintense on DWI and hypointense on ADC maps), distribution (e.g., in the basal ganglia along the lenticulostriate arterial territories) and/or expected progression of diffusion abnormalities over time (i.e., eventual normalization of signal on DWI and increased signal on ADC maps in the long term) (Figure 1A). 

Cryptococcomas do not typically show restricted diffusion [23], but when then do, they tend to show persistent increased hyperintensity DWI, usually with decreased ADC values, over long intervals (weeks to months), which is atypical of ischemia. This is believed to be due to persistent proteinaceous content of the lesions as opposed to transient ischemia-related cytotoxic edema (Figure 1B).

Parenchymal lesions outside the basal ganglia, which are typically referred to as cryptococcomas, were generally hyperintense on T2 and T2-FLAIR, hypointense on T1 and showed variable rim enhancement on contrast enhanced T1-weighted imaging. Findings of meningeal enhancement (Figure 2A), choroid plexus enlargement/abnormal enhancement (Figure 2B) and abnormal ependymal enhancement were determined on contrast-enhanced studies (enhanced T1 and/or enhanced FLAIR images) which were available for 18 patients. Enlarged VR spaces were identified as unusually prominent hyperintense punctate foci on T2 and FLAIR imaging that did not appear as normal variant prominent perivascular spaces (Figure 2C). In a few subjects, borderline prominent T2 hyperintense lesions were seen in the basal ganglia; however, it was not clear whether they represented normal variant prominent perivascular (Virchow–Robin) spaces or pseudocysts. Those were classified as undetermined and were not included in the final tally. Finally, hydrocephalus was described when the ventricular prominence was out of proportion to the sulcal prominence, when the size of the ventricles increased over short time intervals and when transependymal CSF seepage was seen. Entrapment was diagnosed when one or more portions of the ventricle (e.g., occipital or temporal horns) were markedly enlarged compared to the rest of the ventricular system, often in association with transependymal CSF seepage surrounding the entrapped ventricular horn (Figure 2D) and always in association with choroid plexitis.

The quality of the scans was determined as either suboptimal (e.g., motion, lack of contrast, lack of certain sequences) or good (e.g., contrast administration, multiple sequences, follow-up). 

## 3. Results 

### 3.1. Demographics

Demographic details and predisposing conditions for the 29 patients with diagnostic brain MRIs are shown in Table 2 using categories from the REDCap database. Only seven (24%) patients had no underlying disease. The most common underlying condition associated with immunosuppressive therapy was autoimmune disease (28%), followed by solid organ transplantation (21%), and liver disease (17%), a recently reported associated co-morbidity [24]. For patients on immunosuppressive therapy, corticosteroids use was predominant (55%), followed by chemotherapy (31%).

### 3.2. MRI Findings

Table 3 lists the MRI findings of the 29 patients with images reviewed at NIH. Denominators reflect the number of studies in which the assessments could be made. The MRI scans from 15 patients that were not sent to NIH were recorded as “normal” in six (40%) and none were recorded as having hydrocephalus, suggesting that investigators more often submitted abnormal scans. 

Among the MRI scans available for direct evaluation, six were considered of suboptimal quality and for 18 (62% patients), they were performed with contrast. Reads were “within normal limits for age” in three (10%) patients. The most common finding was restricted diffusion foci in the basal ganglia of 11 (38%) subjects and in the cerebella folia, corpus callosum or superficial folia in three (11%), most likely representing arterial infarcts. Seven were judged to have acute/subacute infarcts, mainly in the basal ganglia, based on vascular distribution, location and, when available, signal evolution on subsequent imaging. Four patients in this group were considered to have old lacunar infarcts. In three of the 11 patients, peripheral cortical restricted diffusion foci were seen deep in association with thick meningeal enhancement, which may represent perivascular parenchymal ischemia surrounding inflamed penetrating cortical arteries. Basal ganglia pseudocysts were seen in seven (24%) patients. In the 18 subjects who received contrast, contrast-enhancing parenchymal lesions, referred to as cryptococcomas, were present in the basal ganglia of four (22%) patients and in other locations in four (22%). Meningeal enhancement was seen in 10 patients (56%), choroid plexus enlargement and abnormal enhancement was found in two (11%), and ependymal enhancement in four (24%). (Table 3). 

Hydrocephalus was noted in five of 28 (18%) patients on MRI brain (Table 3) with associated transependymal flow seen in two patients. Four were treated with ventriculoperitoneal shunting. Ventricular entrapment was found in two patients; an obstruction was located at the foramen of Monro in one patient and at the atrium (posterior end of the temporal horn) in the other patient. The unilaterally enlarged ventricular horn in each of those two subjects created marked mass effect and midline shift to the opposite side (features that would preclude a lumbar puncture) and required separate shunting of that part of the ventricular system. Opening pressure was above 25 cm H_2_O in only one of the five hydrocephalus patients, emphasizing the difference between hydrocephalus and increased opening pressure in these patients. 

None of the patients with increased intracranial pressure had evidence of diffuse cerebral edema on imaging. However, increased intracranial pressure was common. Opening pressures on initial lumbar punctures were above 25 cm in 22 (40%) of the 55 patients in whom it was measured. Symptoms and signs of increased intracranial pressure in the absence of increased ventricular size led to repeated lumbar punctures to relieve pressure in 11 of 29 patients (38%), placement of a ventricular drain in one, a lumbar drain in another and ventriculoperitoneal shunts in two patients. Five were given corticosteroids for this indication. One had blindness from high opening pressure, despite serial LPs, corticosteroids and shunting. Corticosteroids had been given for underlying diseases prior to diagnosis in 16 (55%); however, it is unclear if the dose was increased for elevated intracranial pressures. 

## 4. Discussion

Cerebral imaging in our patient population was useful in detecting radiographic abnormalities that were characteristic, though not always specific, of cryptococcal meningitis. On admission, 47% of the CT scans were interpreted as normal, along with 10% of the MRl scans read at NIH and 40% of the MRI scans recorded in the database, consistent with the earlier literature (Table 1) and highlighting the improved sensitivity of MRI compared to CT. Additionally, the data suggest that normal neuroimaging results do not exclude a diagnosis of cryptococcal meningitis. 

Hydrocephalus was detected in five of the 28 patients (18%). Only one of five had an opening pressure on lumbar puncture over 25 cm H_2_O, emphasizing the discordance between hydrocephalus and increased intracranial pressure.

### 4.1. Mass Lesions and Dilated VR Spaces

The most characteristic pathology of cryptococcal meningitis at autopsy is clusters of cryptococci and inflammatory cells extending into the brain along perivascular (Virchow–Robin) spaces, forming what are historically referred to as pseudocysts. Lesions are particularly abundant in the basal ganglia, generally supplied by the lateral and medial lenticulostriate arteries arising from the MCAs and ACAs respectively, supplying the basal ganglia as well as portions of the internal capsules. Pseudocysts were noted in 24% of our patients and did not show contrast enhancement. Pseudocysts likely arise from transit of cryptococci out of blood vessels into perivascular spaces, rather than extension from the meninges [25]. It is important to differentiate those foci from normal variant prominent VR spaces in this location, which can be seen in normal subjects and appear along the lenticulostriate arteries entering the basal ganglia through the anterior perforated substance [26]. Prominent but normal dilated VR spaces show increased signal on T2 and decreased signal on FLAIR, consistent with CSF signal being suppressed on FLAIR. Since pseudocysts are filled with cryptococci, they often retain signal intensity on FLAIR imaging, due to the gelatinous consistency of their contents. Occasionally, this can result in restricted diffusion.

Parenchymal brain masses that contain numerous cryptococci surrounded by gelatinous capsule material, lymphocytes and histiocytes have been termed cryptococcomas. The pathology of brains at autopsy has found no clear distinction between the clustered, punctate masses called pseudocysts and the larger, solid masses called cryptococcomas [14]. Cryptococcomas were seen in our patient population, with the largest lesions being in the basal ganglia. Depending on the inflammatory response, cryptococcomas may or may not have rim enhancement on contrast-enhanced T1 imaging [23].

Another characteristic autopsy finding is inflammation and numerous cryptococci in the meninges. This inflammatory response resulted in meningeal enhancement in 56% of our patients who received contrast. These values were somewhat lower than the 45–71% recorded in the literature for meningeal enhancement in non-HIV patients (Table 1). Our numbers, however, are likely an underestimate since enhanced FLAIR images were only obtained in three subjects. Enhanced T1-weighted imaging is less sensitive than enhanced FLAIR imaging for detection of meningeal enhancement [27] and as such is recommended in patients with suspected meningeal disease in general, and more specifically in patients cryptococcal infection. 

Enhancement of the ventricular lining (ependyma) was seen in 24% while choroid plexus enlargement and enhancement (choroid plexitis) was seen in 11%. Ependymitis and choroid plexitis led to the entrapment of portions of the lateral ventricles in two patients, likely due to adhesions, resulting in mass effect that required neurosurgical drainage. 

### 4.2. Infarcts and Atrophy

The incidence of infarcts in our patients is in line with prior reports (Table 1). Infarcts diagnosed by MRI are often multiple and do not always present with stroke-like symptoms [28]. The etiology of these lesions is complex and requires the correlation of imaging with pathology [29]. Age-related atherosclerosis and vasculitis from underlying diseases, such as systemic lupus erythematosus in one of our patients, may have contributed to the incidence of the infarcts. Inflammation of cerebral vessels, i.e., vasculitis, is not mentioned in pathology reports of cryptococcal meningitis [30,31]. In published reports, CT angiography and MR angiography (MRA) have either shown normal or narrowed cerebral arteries in cryptococcal meningitis patients with infarcts [32,33,34] Most of those infarcts were seen in the basal ganglia, along the distribution of the lenticulostriate arteries. 

In some of our patients, foci of restricted diffusion were also seen involving superficial cortical layers deep to meningeal enhancement foci. We believe those might be due to cytotoxic edema in cortical regions where the inflammatory/infectious changes in the adjacent meninges extend along the penetrating arteries, causing focal ischemic changes. Restricted diffusion foci were also seen in locations that are unusually affected by vascular disease such as the corpus callosum. The latter could possibly reflect cryptococcomas, although a detailed longitudinal analysis of a well-defined cohort remains necessary to better differentiate vascular etiologies from those associated with infectious/inflammatory processes. Another imaging feature, diffuse cerebral atrophy, is included in some reports of cryptococcal meningitis in patients with HIV, where HIV may be a major contributor, but has been infrequently commented upon in non-HIV patients with cryptococcal meningitis (Table 1). It may be relevant that there is a report of diffuse cerebral atrophy in children as judged by CT in nine of 11 immunocompetent patients with *Cryptococcus gattii* meningitis [35]. We did not attempt to distinguish atrophy from age related changes.

### 4.3. Clinical Correlates

No imaging correlates for the reported decreased vision or cranial nerve palsies were found. Visual loss in cryptococcal meningitis is often considered to originate in the optic nerve, disc or chiasm, areas not specifically imaged in our study [36]. Although increased diameter of the optic sheath, measured 3 mm behind the globe, has been reported to correlate with increased intracranial pressure, the specific imaging requirements (dedicated imaging of the orbit with and without fat suppression) were not met in our study [37]. To detect MRI abnormalities associated with cranial nerve palsies, additional imaging techniques may be needed, as we have reported with decreased auditory function in cryptococcal meningitis [38]. 

We did not find a correlation between MOCA scores and infarcts or other lesions with restricted diffusion, but the small numbers may have limited our ability to detect such relationships. Elsewhere, MRI has been used to show that infarctions have a negative impact on outcome [29]. Diffusion tensor imaging has found correlates between location of white matter lesions and decreased function on neuropsychological testing [12,39]. Cryptococcal lesions in the basal ganglia or centrum ovale have been correlated with poor neurologic outcome [20].

### 4.4. Limitations

Several factors limit our imaging studies. First, MRI images were available for review for only 29 patients, limiting our sample size and hence the representative conclusions from this study. It is also likely that patients with proven CM in the CINCH study whose MRI images were not submitted for review had fewer abnormalities on imaging. This would inflate the percentages of abnormalities in the 29 patients whose images are reported here in detail. Another limitation of the MRI data is that 38% of the MRI scans were not contrast-enhanced, perhaps due to azotemia or time limitations for ill patients. Three patients had their first MRI more than 15 days after diagnosis. Patient motion degraded the MRI images in two patients. MR scanners also varied in their quality (1.5 versus 3T for example) and different imaging sequences/parameters were obtained in different centers. The paucity of follow-up MRI also prevented accurate assessment of progression of the abnormalities over time. However, this experience reported here may be representative of the usual experience when imaging acutely ill patients with cryptococcal meningitis. These patients are often confused or lethargic on admission and cannot lie still in a scanner for prolonged periods. Improved imaging options may thus benefit management of cryptococcal meningitis.

## Figures and Tables

**Figure 1 jof-09-00594-f001:**
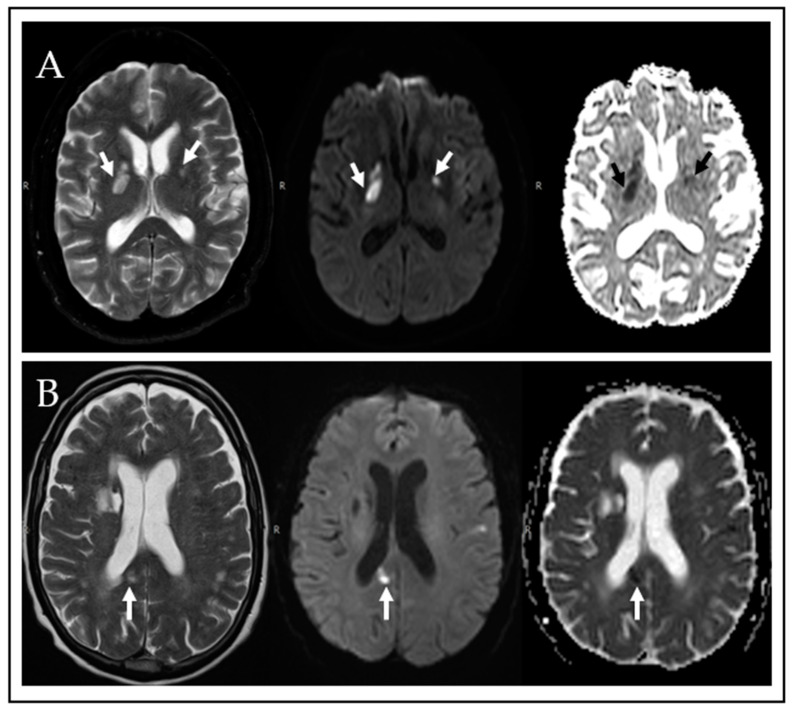
MRI findings in CM including ischemic infarcts or cryptococcal-space-occupying lesions. (**A**) Bilateral basal ganglia acute infracts showing high signal on T2 weighted images (left image), restricted diffusion with high signal on DWI (mid image) and low signal on ADC maps (right image). (**B**) Image of likely cryptococcoma in the splenium of the corpus callosum (see arrows), showing high signal on T2 (left image), restricted diffusion with high signal on DWI (mid image) and low signal on ADC maps (right signal). While acute infracts show expected progression of ischemic lesions on follow-up, cryptococcomas are expected to show persistent restricted diffusion over weeks to months.

**Figure 2 jof-09-00594-f002:**
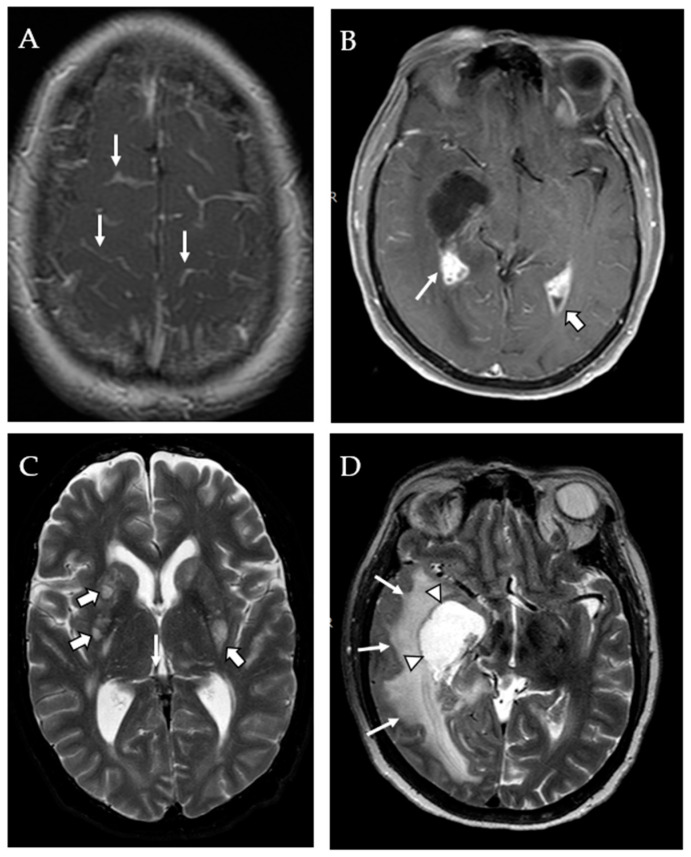
MRI findings in CM including intracranial enhancement and hydrocephalus. (**A**) Extensive meningeal enhancement seen on postcontrast T1-weighted images (thin white arrows). (**B**) Choroid plexus enlargement and abnormal enhancement (thin white arrow) and abnormal ependymal enhancement (thick white arrow) seen on postcontrast T1-weighted images. (**C**) Abnormally enlarged Virchow–Robin spaces on T2-weighted images suggesting accumulation of cryptococci (pseudocysts) (arrows). (**D**) Right temporal horn entrapment (arrow heads) with transependymal CSF seepage and secondary mass effect (thin white arrows).

**Table 2 jof-09-00594-t002:** Demographics and predisposing conditions in 29 patients with diagnostic MRIs.

Age, Median Years (IQR)	61 (53–70)
Sex, female	11 (38%)
Male	18(62%)
Race,	
Caucasian	23(79%)
African American	4 (14%)
Asian	2 (7%)
Documented cryptococcal lung infection	9 (31%)
Underlying Disease	
Solid organ transplantation	6 (21%)
Solid Tumor	2(7%)
Hematologic Malignancy *	3 (11%)
Liver Disease	5(17%)
Liver Cirrhosis	4(14%)
Hepatitis	2 (7%)
Autoimmune Syndromes ^^^	8 (28%)
Primary immunodeficiency ^+^	3 (10%)
Diabetes mellitus	10(34%)
None	7(24%)
Immunosuppressive medications	
Glucocorticoids	16 (55%)
Cytotoxic chemotherapy	9 (31%)
Calcineurin/mTOR inhibitors **	5 (19%)
Antimetabolites **	6 (22%)
Targeted antibodies **	1(4%)

* missing data with denominator being 28 patients; ** missing data with denominator being 27 patients. ^^^ Autoimmune syndromes included rheumatoid arthritis, psoriasis, polyarteritis nodosa, systemic lupus erythematosus, idiopathic chronic thrombocytopenia, multiple sclerosis and myasthenia gravis. ^+^ Of 3 patients with primary immunodeficiency, two had idiopathic chronic lymphopenia and one had monoclonal gammopathy of undetermined significance.

**Table 3 jof-09-00594-t003:** Results of imaging studies.

Study	Result	# Cases	Evaluable	%
CT	Normal	24	51	47%
MRI	Normal	3	29	10%
	Hydrocephalus	5	28	18%
	transependymal flow	2	28	7%
	lesions with restricted diffusion (infarcts)			
	basal ganglia	11	29	38%
	elsewhere *	3	28	11%
	pseudocysts in basal ganglia	7	29	24%
	gadolinium contrast	18	29	62%
	parenchymal enhancing lesions (“cryptococcomas”)			
	basal ganglia	4	18	22%
	elsewhere	4	18	22%
	meningeal enhancement	10	18	56%
	choroid plexus enhancement	2	18	11%
	ependymal enhancement	4	17	24%

* Cerebellar folia, corpus callosum, superficial cortex. ^#^ Refers to number of cases.

## Data Availability

The data presented in this study are available on request from the corresponding author. The data are not publicly available due to privacy restrictions.

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
