# Peer review of "Neuroimaging of Cryptococcal Meningitis in Patients without Human Immunodeficiency Virus: Data from a Multi-Center Cohort Study"

_jof, 2023, doi:10.3390/jof9050594_

Round 1

Reviewer 1 Report

The manuscript attempted to investigate neuroimaging of CM in patients without HIV. In this study, CT was performed in 51 and MRI in 44. MRI results are reported for the images read at NIH for 29 of the 44 patients. The authors reported MRI characteristics of non-HIV CM include hydrocephalus, meningeal and ependymal enhancement and basal ganglia lesions. However, the manuscript is not suitable to publication in current form and need to be substantially revised, because there are several limitations of this study, as shown below:

Major points

1. Due to the small sample size, there is too little information to draw representative conclusions.

2. According to authors result, MOCA scores were available for 3 with hydrocephalus. How to conclude that MOCA is not correlated to infarcts or other lesions with restricted diffusion? In addition, please provide detailed  statistical data.

3. In the first paragraph in the Discussion section, it was highlighted that MRI is more sensitive than CT, but there was no comparison between MRI and CT in the manuscript.   

4. The CT findings were not described in the results. 24 of 51 (47%) are normal, what are the abnormalities in the rest of the population? What is the consistency between CT and MRI?

5. In Results, the subtitle needs to be modified. The 3.2 subtitle is MRI findings, while the content described in the following subtitles are mainly MRI findings.

6. Among HIV-negative patients, some may develop post-infectious inflammatory syndrome (PIIRS)/IRIS-like syndrome after effective anti-fungal treatment. What percentage of these patients have PIIRS/IRIS-like syndrome? Could the authors provide neuroimaging data on PIIRS/IRIS-like syndrome?

Minor points

1. The word meningoencephalitis is used in the title, but the word meningitis is used in the abstract and main text. It needs to be unified. Suggest changing the word to meningitis.  

2. Please delete the Hematopoietic stem cell transplantation column in Table 2.

3. In Table 2, could you describe in more detail the specific diseases in the Autoimmune Syndromes and Primary immunodeficiency columns?

Author Response

All line numbers referred to are in the tracked and not clean version provided to the editor 

Major points

  1. Due to the small sample size, there is too little information to draw representative conclusions.

A - We now point this out this limitation on lines 317-318.We recognize the small sample size as a limitation

  1. According to authors’ result, MOCA scores were available for 3 with hydrocephalus. How to conclude that MOCA is not correlated to infarcts or other lesions with restricted diffusion? In addition, please provide detailed  statistical data.

A- Statements about MOCA scores of three hydrocephalic pattients on lines 201-205 were removed because the sample size was small.

  1. In the first paragraph in the Discussion section, it was highlighted that MRI is more sensitive than CT, but there was no comparison between MRI and CT in the manuscript.  

A- Table 3 compares the incidence of normal CT and MRI. Also see lines 91-93, under section 2.1 (Available Imaging studies), we mention that with the exception of normal results, the imaging results of CTs in the REDCap database are not included due to the incompleteness of data. This is the same reason for not comparing results of the 29 MRIs reviewed for the paper with results recorded by investigators in the REDCap database.

  1. The CT findings were not described in the results. 24 of 51 (47%) are normal, what are the abnormalities in the rest of the population? What is the consistency between CT and MRI? 

A- As explained above and under section 2.1, CT abnormalities could not be commented upon due to the incompleteness of data entered into the REDCap database.

  1. In Results, the subtitle needs to be modified. The 3.2 subtitle is MRI findings, while the content described in the following subtitles are mainly MRI findings.

A- To avoid confusion, we have removed the subtitles of hydrocephalus and increased intracranial pressure and imaging so that all MRI findings are under section 3.2

  1. Among HIV-negative patients, some may develop post-infectious inflammatory syndrome (PIIRS)/IRIS-like syndrome after effective anti-fungal treatment. What percentage of these patients have PIIRS/IRIS-like syndrome? Could the authors provide neuroimaging data on PIIRS/IRIS-like syndrome?

A- Followup neuroimaging was not a part of the study and not often performed so we could not evaluate inflammation during therapy. The percentage of PIIRS in previously healthy patients is roughly 30% but in HIV-negative patients with cryptococcal meningitis and a wide variety of predisposing conditions, the percentage is unknown at present. In collaboration with Hopkins, we are in the process of designing a multi-center study to answer this question. Neuro-imaging data on PIIRS in previously healthy patients can be found in our previously published study in CID (PMC ID 8563180) under the “Radiological Outcomes” section. The most common findings were meningeal enhancement and hydrocephalus followed by parenchymal enhancement.

Minor points

  1. The word meningoencephalitis is used in the title, but the word meningitis is used in the abstract and main text. It needs to be unified. Suggest changing the word to meningitis.  

Wording in the title and elsewhere has been changed to “meningitis”

  1. Please delete the “Hematopoietic stem cell transplantation” column in Table 2.

This had been deleted.

  1. In Table 2, could you describe in more detail the specific diseases in the “Autoimmune Syndromes” and “Primary immunodeficiency” columns?

A legend has been added to table 2. Of the 8 patients who had autoimmune syndromes in our group, 2 had rheumatoid arthritis, 1 had psoriasis, 1 had polyarteritis nodosa, 1 had lupus, 1 had idiopathic chronic thrombocytopenia, I had multiple sclerosis and 1 had myasthenia gravis. Of the 3 patients with primary immune deficiency, 2 had idiopathic chronic lymphopenia and 1 had monoclonal gammopathy of unknown significance.

Reviewer 2 Report

This  is well prepared report that analyzed the brain CT and MRI data for non-HIV patients with CM. CM is a significant fungal disease with very high mortality. Accurate diagnosis is highly important for patient treatment., This study found the MRI is more sensitive than CT , yet optimized imaging still is needed to improve the diagnosis accuracy. This report is clinical relevant and hight significant.

Author Response

Thanks you for the great review. 

Reviewer 3 Report

This interesting study reports neuroimaging findings in non-HIV individuals with cryptococcal meningitis. Some aspects need to be improved before the article is accepted for publication. 

1. In Methods Section, enlarged Virchow-Robin spaces and pseudocysts are presented as synonyms. However, both presentations can be differentiated accotding to their size. This point need to be clarified and incorporated throughout the article. In clinical practice, pseudocysts need a more prolongued induction therapy, similar to cryptococcomas. 

2. In Results Section, the authors should show the main neuroimaging features of the 7 patients without underlying diseases. For example, these patients had a more "inflammatory profile" in neuroimaging?. They had more cryptococcomas or pseudocysts?.

3. In Discussion section was wrote: "...along perivascular (Virchow-Robin) spaces, forming what are historically referred to as pseudocysts and, when larger, cryptococcomas." The authors need to clarify the histopathological difference between pseudocysts and cryptococcomas and correct this sentence. 

4. In Introduction and Results sections, the presentations of tables needs to be improved. 

5. Despite having only 6 patients with neuroimaging follow-up, I consider a brief description of them is important. What was the post-treatment evolution of pseudocysts or cryptococcomas?. 

Author Response

All line numbers referred to are in the tracked version of the revised manuscript which has been emailed to the editor.

This interesting study reports neuroimaging findings in non-HIV individuals with cryptococcal meningitis. Some aspects need to be improved before the article is accepted for publication. 

  1. In Methods Section, enlarged Virchow-Robin spaces and pseudocysts are presented as synonyms. However, both presentations can be differentiated accotding to their size. This point need to be clarified and incorporated throughout the article. In clinical practice, pseudocysts need a more prolongued induction therapy, similar to cryptococcomas. 

A- Please see additional clarification on lines 61-62, 128 and 251-257.

  1. In Results Section, the authors should show the main neuroimaging features of the 7 patients without underlying diseases. For example, these patients had a more "inflammatory profile" in neuroimaging?. They had more cryptococcomas or pseudocysts?.

A- We were surprised that there was no  obvious difference in the images of the seven immunocompetent patients vs the other 22 patients but didn’t comment because numbers were small. For the reviewer’s interest,  signs of inflammation shown by contrast enhancement in the 18 imaged after gadolinium contrast were as follows, listing immunocompetent vs the other patients: meninges 3/6 vs 7/12, ependyma 2/6 vs 2/12 and choroid 1/6 vs 2/12.

  1. In Discussion section was wrote: "...along perivascular (Virchow-Robin) spaces, forming what are historically referred to as pseudocysts and, when larger, cryptococcomas." The authors need to clarify the histopathological difference between pseudocysts and cryptococcomas and correct this sentence. 

A- Please see added lines 251-257 in the Discussion. We explain that on autopsy specimens, pseudocyts are gelatinous masses of cryptococci that cause enlargement of the Virchow Robin spaces and it is the confluence of these spaces that results in a “cystic appearance” whereas cryptococcomas are larger parenchymal masses. We provide a new reference that correlates pathology with imaging.

In Introduction and Results sections, the presentations of tables needs to be improved. 

A- Formatting was done by publisher.

  1. Despite having only 6 patients with neuroimaging follow-up, I consider a brief description of them is important. What was the post-treatment evolution of pseudocysts or cryptococcomas?. 

A- Our neuroradiologist does not feel she can comment on resolution of cryptococcoma or pseudocysts because data are so limited. Only two followup MRIs were done with contrast. Timing of the followup MRIs tended be very early (two weeks in 4 patients) or beyond 4 months and only a single followup MRI was done in four patients. For the reviewer’s interest, of the 4 patients who had lesions on initial imaging, 2 had resolution of lesions suggesting infarcts on MRI done 2 weeks later, 1 had worsened cryptococcomas on MRI brain done 1 month later and one in whom enlarged perivascular spaces were seen initially, follow up MRI was not done with DWI or T2 weighted images to reassess the spaces.

Round 2

Reviewer 1 Report

The manuscript has been sufficiently improved to warrant publication in JoF. 

Reviewer 3 Report

The current version of the article is adequate for publication.